# Multiscale Variability of Hydrological Responses in Urbanizing Watershed

**Urmila R. Panikkar** [1], **Roshan Srivastav** [1,*] and **Ankur Srivastava** [2]

1 Department of Civil and Environmental Engineering, Indian Institute of Technology Tirupati, Yerpedu, Tirupati 517619, India
2 Faculty of Science, University of Technology Sydney, Sydney, NSW 2007, Australia
* Correspondence: roshan@iittp.ac.in

**Abstract:** Anthropically-induced land-use/land cover (LULC) changes create an imbalance between water and energy fluxes by affecting rainfall-runoff partitioning. This alters the catchment's flow regime, generating increased highs and reduced low flows, triggering socio-economic and environmental damages. The focus of this study is two-fold (i) to quantify the hydrological changes induced in the urbanizing watershed and (ii) to analyze changes in streamflow variability and generation of extremes (high- and low-flow), using the soil and water assessment tool (SWAT) for Peachtree Creek, USA. The results indicate that the change in LULC significantly influences the availability of soil moisture, ET, and contribution to groundwater flow. It is observed that the variations in these processes regulate the water availability from the surface and sub-surface sources, thus affecting the generation of extreme flows. The spatio-temporal analysis, in response to LULC changes, indicates that (i) urbanization significantly affects baseflow, and its variability depends on the degree of urbanization and the predominant land-use class of the subwatersheds, and (ii) the seasonal variations in the baseflow contribution to the streams depend on ET and the timing and magnitude of groundwater outflow to streams. These variations in ET and groundwater lead to water excess/deficit regions, thus increasing the susceptibility to floods during heavy precipitation events and reducing the reliability of streams during dry periods. Thus, in an urbanizing watershed, the hydrological regime of the watershed may not always be a function of changes in the surface runoff, but will be modified by ET and groundwater dynamics. Further, the study shows that the changes in model parameters can provide insight into the implications of LULC changes on hydrological processes and flow regimes. Evaluating the implications on the basin water balance is paramount for deriving any management operations and restoration activities. The study also outlines the significance of analyzing the spatial and temporal scale streamflow variations for managing water resources to reduce damage to lives and properties.

**Keywords:** extremes; urbanization; anthropic activities; land-use/land-cover; streamflow; hydrologic parameters





## 1. Introduction

The water-related disasters are associated with a high degree of randomness and uncertainty, triggering substantial socioeconomic and environmental damages. For example, in India, out of all the natural disasters, 40% of deaths are due to floods alone, and between 1980 and 2017, the economic loss from floods accounted for about $58.7 billion (Emergency Events Database, EM-DAT 2018). Recent years have witnessed a rise in the frequency, magnitude, and intensity of these events, which are direct consequences of meteorological forcings, mostly accelerated by anthropic-induced changes [1,2]. Zhang [2] studied the influence of anthropogenic forces on precipitation trends and suggested an increase in global mean precipitation, changes in the latitudinal pattern of precipitation (i.e., an increase in precipitation in high latitudes and a decrease in sub-tropical latitudes),

and a shift in the position of intertropical convergence zone (ITCZ). The changes in land-use patterns, especially in urbanizing watersheds, cause substantial variations in temperature and moisture, which, in turn, alters the hydrological regime of the catchment [3,4]. This results in increased high flows (vulnerable to flooding during heavy precipitation events) and reduced low flows (affect water availability as well as water quality) [5]. In addition, an increase in water demand further affects the low flow conditions. Thus, there is a need to identify the changes in high- and low-flow conditions of the watershed for sustainable management of water resources.

Anthropic-induced land-use/land-cover (LULC) changes create an imbalance between water and energy fluxes, thus affecting rainfall-runoff partitioning by triggering variations in evapotranspiration (ET) rates, groundwater recharge, and soil water. It is observed that an increase in impervious fraction leads to water accumulation in the upper soil depths, thus increasing surface runoff and ET [3]. Vaibhav [4] observed an increase in annual surface runoff by approximately 45%, due to an increase in the urban area and a reduction in forest and cropland. Cong [6] conducted a study to estimate ET in plain, mountainous, urban, and sub-urban areas using the surface energy balance system (SEBS) model. The results indicated that ET rates is highest in urban areas and lowest in mountainous areas. Siddik [7] observed a reduction in groundwater recharge by 17.1 mm/year, due to an increase in the built-up area and bare soil and a reduction in open water sources, due to meandering. Galiano [8] indicated that the conversion of agricultural land to built-up areas altered the flood characteristics of the catchment and increased flood peak discharge and volume by 24 and 26%, respectively. Thus, the hydrological response of the watershed is strongly correlated to changes in LULC patterns.

LULC changes in a watershed over the years also show sub-basin scale diversification. Garg [4] studied the impact of different LULC scenarios on basin water balance components (WBCs). It was observed that the upstream part of the basin showed an increasing trend in all WBCs, whereas the lower part of the basin showed a decreasing trend, indicating spatial variation in hydrological components. Pumo [3] indicated that rainfall-runoff response could significantly change with spatial scale. The study suggested that, at a larger scale, alterations in both runoff and evapotranspiration were less evident, with imposed changes in LULC. Further, the LULC changes also influence the temporal behavior of WBCs [9]. Li [10] observed that, with the expansion of agricultural land, there was a sharp decline in streamflow during the wet season, due to reduced groundwater contribution, whereas a slight increase in streamflow during the dry season was due to a decrease in soil water. Serur [11] indicated significant variations in the temporal distribution of WBCs and streamflow, with an increase in agriculture land and urban areas. An increase in wet season streamflow (due to higher surface runoff) and a reduction in dry season streamflow were observed (due to reduced baseflow and higher ET). Chen [12] analyzed the impact of desertification on soil moisture content due to the clearing of vegetation. The study concluded that land-cover changes alter porosity, soil structure matrix, and surface energy, leading to variations in the spatio-temporal distribution of soil water. Therefore, identifying the multi-scale variability in the hydrological response of watersheds helps policymakers in decision-making.

Soil water assessment tool (SWAT), developed by the US Department of Agriculture (USDA), is a physical-based semi-distributed hydrologic model. Several studies have reported the use of SWAT to identify the impact of various management activities on the surface flows, groundwater storage, and also in the simulation of hydro-climatic extremes [13–15]. The SWAT model is also adopted to examine the change in hydrological processes in response to LULC alterations. Palamuleni [16] investigated the role of land cover changes of the Upper Shire River, Malawi, in the degradation of flow regimes using the SWAT model. The study analyzed the trends in land cover change between 1989 and 2002, and the simulations indicated an increase in peak flows and faster time of travel with land cover changes. Zhu [17] assessed the long-term hydrological impact in the Little River Watershed, Tennessee, using the SWAT model and LULC change from 1984–2010.

The model calibrated and validated for the observed streamflow of 2010 and simulated for different LULC change scenarios indicated an increase in streamflow with urban expansion. Munoth [18] studied the impact of LULC change on runoff and sediment yield in the Upper Tapi River sub-basin, India. The LULC maps corresponding to the years 1975, 1990, 2000, and 2016, with the corresponding climate data, were used to develop four different models and were calibrated separately. The model simulations by changing the LULC maps with their calibrated parameters revealed an increase in runoff, water yield, and sediment yield, with agriculture expansion and loss of forest cover. Mengistu [19] calibrated the simulation of each reference LULC period (2000, 2010, and 2020) obtained from the SWAT model to understand the implications of LULC changes in Gilgel Gibe, an East African watershed. The study indicated an increase in surface runoff and a decrease in groundwater recharge, with the expansion of agricultural and grasslands and loss of forest. Ref. [20] used the SWAT model to identify the effect of LULC changes on the natural flow of the Ramganga River Basin, India, for a period of 43 years. The model simulation, with Nash–Sutcliffe efficiency as a performance indicator, gave a good correlation between observed and simulated flows and indicated an increase in the average natural flows.

Despite numerous studies on the implications of LULC changes on watershed hydrology, the following aspects were not given much attention: (i) how variations in model parameters related to soil, evaporation, and groundwater, as a consequence of LULC changes and the influence of spatial and temporal changes in streamflow, and (ii) the interplay of these parameters, the land-use pattern, and the intra-annual variations in climate in the generation of extreme flows. Further, the studies also neglected to examine the nature of hydrological response causing spatio-temporal variations in streamflow and generation of extremes [3,4,7]. Identifying the sub-basin scale change in land-use pattern and streamflow helps in locating water-stressed/excess regions and limiting the development activities. Temporal variations in streamflow help in identifying the flow regime of the basin, analyzing the occurrence of extreme events and developing an optimal framework for the seasonal allocation of water resources. One way of understanding the mechanism involved in spatio-temporal variability of streamflow is through model parameters, as it reflects changes in water balance components (WBCs) in response to LULC alterations. The focus of this paper is to study the effect of anthropic-induced LULC changes on extreme streamflow generation using the soil and water assessment tool (SWAT). The specific objectives are to (i) identify the variations in model parameters and corresponding implications on WBCs in response to LULC changes, (ii) analyze spatial and temporal variability in streamflow, and (ii) quantify the effect on the generation of extreme (high- and low-flow). The study is conducted in the Peachtree Creek watershed, Atlanta, Georgia, having a considerable degree of urbanization.

The paper is organized as follows. The research methodology is outlined in Section 2. Section 3 presents the observation obtained by SWAT modeling and discussions on critical results. Section 4 contains a summary and conclusion, followed by references.

## 2. Materials and Methods

### 2.1. Study Area

The Peachtree Creek study basin (Figure 1), located in the Piedmont Plateau physiographic province, Atlanta, Georgia, USA, encompasses an area of 211.84 km², with an altitude of 299 m above sea level. Peachtree Creek, a tributary of Chattahoochee River, is situated at a latitude of 33°49′10″ N and a longitude of 84°24′28″ W. The Piedmont region is known for its rocky, clay-rich soils and its abundance of quartz, feldspar, and mica, which are common minerals in the region. The area may also have some sedimentary rocks, such as sandstone and limestone, which were formed from the accumulation of sediment over time [21]. In addition to these geologic features, the watershed also has natural features, such as streams, rivers, and wetlands, which are important components of the local ecosystem.

The urbanizing watershed of Peachtree covers a major portion of metropolitan Atlanta and has a population density greater than 2400 people/km² (Atlanta Regional Commission, 200b). The undeveloped areas are mainly forested and hilly [22]. The dominant land-use class of the basin is urban residential (60%). The soil in the area is predominantly sandy loam, belonging to hydrologic soil group B, based on SCS-CN method [23]. Fine-grained silty clay is found along the banks, whereas the bed consists of medium sand. The groundwater level exhibits seasonal variations, ranging from 2.5 to 1 m below the land surface, with the highest water table occurring during winter, and the lowest occurring during the growing season in summer [22].

The region experiences a humid and sub-tropical climate and an average annual rainfall of 1200 mm. The maximum and minimum rainfall are received in February and October, respectively. The yearly average temperature is 16.5 °C—July is the hottest month, with an average high temperature of 32 °C, and January is the coldest month, with an average low temperature of 2 °C. Streamflow generation involves a contribution from surface runoff, shallow soil layer, hillslope groundwater, and deeper groundwater within the regolith [22]. In the watershed, 66% of the total annual runoff is generated from stormwater runoff [22]. The increasing population growth and rapid pace of urbanization have posed heavy pressure on the water resources in the Piedmont province [24]. Moreover, the small urban stream area of Peachtree Creek and the increase in impervious fraction causes greater velocity and volume of surface runoff, i.e., short-term flood peaks tend to increase [25]. Thus, it is vital to understand how anthropic-induced LULC changes alter the watershed's hydrological processes affect streamflow's quantity and reliability.

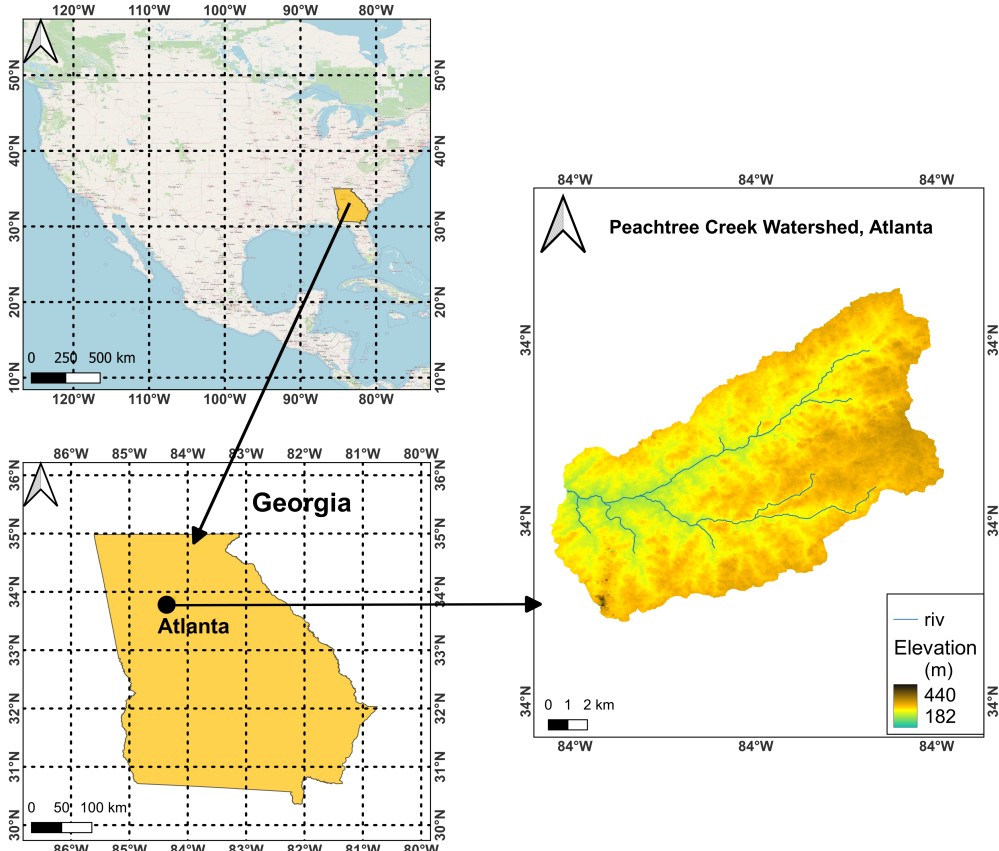

**Figure 1.** Map showing Peachtree creek watershed, Atlanta, Georgia, USA.

### 2.2. Data Sources

The input data for SWAT includes digital elevation model (DEM), land-use and land-cover (LULC) maps, soil properties, and climate data. The source of the input datasets for SWAT has a significant impact on streamflow prediction [26]. Chordia [26] indicated that, for the Peachtree Creek watershed, the Advanced Spaceborne Thermal Emission and Reflection Radiometer (ASTER) DEM, National Land Cover Database (NLCD) LULC, and Tropical Precipitation Measuring Mission (TRMM) datasets gave good Nash–Sutcliffe coefficient (NSE) values in the simulation of streamflow.

In this study, (i) an ASTER DEM of 30 m resolution was obtained from the USGS earth explorer website (https://earthexplorer.usgs.gov/ (accessed on 20 October 2022)); (ii) LULC data for the years 2011 and 2019 were obtained at 30 m resolution from the NLCD database derived from Landsat-based imagery (https://www.mrlc.gov/ (accessed on 20 October 2022)); (iii) soil data were obtained from Soil Survey Geographic Database (SSURGO), developed by USDA-NRCS (Natural Resources Conservation Agency); (iv) daily climate data for precipitation, minimum and maximum temperature, solar radiation, wind speed, and relative humidity were collected from 2007 to 2019. The remotely sensed precipitation estimates of 0.25° × 0.25° were downloaded from TRMM dataset (https://trmm.gsfc.nasa.gov/ (accessed on 20 October 2022)) for years 2007–2019. For daily temperature, wind speed, solar radiation, and relative humidity, climate forecast system reanalysis (CFSR) data of 30 km resolution were obtained from http://globalweather.tamu.edu/ (accessed on 20 October 2022); and (v) flow measurements for the study region are obtained from the USGS Georgia District Database from the USGS monitoring station (02336300) in Peachtree Creek, Atlanta.

### 2.3. SWAT Model

SWAT is a continuous, semi-distributed model that operates on a daily time scale. It was developed to study the impact of various management activities on water, agriculture chemical yields, and groundwater. The hydrological processes in SWAT, driven by water balance, include canopy storage, evapotranspiration, surface runoff, return flow, lateral flow, and percolation to shallow and deep aquifers. The hydrologic simulation is based on the water balance equation, given by:

$$[SW]_t = [SW]_0 + \sum_{i-1}^{t}[R - Q - ET - P - GW] \tag{1}$$

$SW_t$ is the final soil water content on day i ($mmH_2O$), $SW_0$ is the initial soil water content on day i ($mmH_2O$), $t$ is time (days), $R$ is the amount of precipitation on day $i$ ($mmH_2O$), $ET$ is the amount of evapotranspiration on day $i$ ($mmH_2O$), $P$ is the amount of percolation and bypass flow exiting the soil profile bottom on day $i$ ($mmH_2O$), $Q$ is the amount of surface runoff on day $i$ ($mmH_2O$), and $GW$ is the amount of return flow on day $i$ ($mmH_2O$). In SWAT, the land area in a sub-basin is divided into hydrologic response units (HRUs) that possess a unique combination of land-use, soil characteristics, and slope classes. HRU analysis in SWAT helps to evaluate the spatial variation of water balance components of the basin. Equation (1) is applied for each HRU, and the amount of water for each water balance component is then summed at the sub-basin level, which is then connected to stimulate water transport through stream network [27]. In SWAT, surface runoff is calculated using soil conservation service curve number method (SCS). Penman–Monteith, Hargreaves, and Priestley–Taylor methods are available for predicting evapotranspiration. Water that seeps below the soil profile is partitioned between shallow and deep aquifers. Evapotranspiration from deep-rooted plants and return flow to the stream occur from the shallow aquifer. Recharge to the deep aquifer is assumed to be lost from the system [28].

### 2.4. Selection of SWAT Parameters

LULC changes can alter surface runoff, ET, overland flow velocities, and infiltration rates. Hence, the parameters controlling these processes change with LULC scenarios. For this study, 15 parameters were selected based on their relevance to hydrological processes, study area, and literature reviews [17,28–31]. The parameter selected were studied for their influence on streamflow.

The parameters initial SCS curve number for moisture condition II (CN2) and surface runoff lag time (SURLAG) were used to signify the magnitude and timing of surface runoff. CN2 determines the runoff depth based on soil permeability, land-use/land-cover, and antecedent moisture conditions. A decrease in CN2 indicated reduced surface runoff generation and increased water availability for baseflow and groundwater percolations. SURLAG represents the basin response by controlling the fraction of surface runoff reaching the channel on a specific day and overland flow velocities on the basin surface. A decrease in SURLAG indicates a reduction in the amount of water reaching the sub-basin outlet [32]. These parameters describe the effect of LULC changes on surface runoff characteristics. The available soil water capacity (SOL_AWC) and soil compensation factor (ESCO) control the evapotranspiration and percolation to groundwater, as well as the occurrence of surface and lateral flows [28]. SOL_AWC is obtained by deducting water content at the permanent wilting point from the field capacity. An increase in SOL_AWC indicates higher water holding capacity of the soil, increased ET, and thus, decreased streamflow. The soil evaporation compensation factor (ESCO) indicates evaporation from a soil layer, which is the difference in the evaporative demand of the upper and lower boundaries of the soil layer. An increase in ESCO means that soil evaporative and plant demand can be met from the upper layers, thus minimizing the water dawn from deeper layers. Thus, increasing streamflow through reduced ET. The parameters SOL_AWC and ESCO reflect the soil moisture variability with LULC alterations, thus affecting the surface and sub-surface distribution of precipitation.

Loss of native vegetation cover and land management practices, such as tillage, alteration of drainage networks, etc., influences the hydro-physical properties of soil. This consequently alters the infiltration rate and groundwater recharge [33]. Therefore, the model parameters governing sub-surface processes will exhibit variations with LULC changes. The groundwater parameters such as groundwater "revap" coefficient (GW_REVAP), threshold depth of water in the shallow aquifer required for return flow to occur to the stream (GWQMN), and threshold depth of water in the shallow aquifer for "revap" to occur (REVAPMN) controls the movement of water from shallow aquifer to unsaturated zone and the occurrence of baseflow. Low values of GWQMN indicate an increase in the return flow from the shallow aquifer. A higher value of GW_REVAP and a low value of REVAPMN suggest water movement to rootzone for 'revap' and, hence, reduced baseflow. These parameters are significant in watersheds subject to the removal of deep-rooted vegetation. Another parameter that governs the amount of baseflow contributed to streams is the deep aquifer percolation fraction (RCHRG_DP). It controls the amount of water percolating from the root zone to the deep aquifer and, thus, separates shallow and deep aquifer recharges [29]. An increase in the value of RCHRG_DP indicates decreased baseflow and increased deep aquifer recharges. The groundwater time delay (GW_DELAY) and baseflow recession constant (ALPHA_BF) control the timing of sub-surface processes. GW_DELAY represents the delay time for recharging the shallow aquifer, and an increase in the delay factor corresponds to slow recharge processes [28]. ALPHA_BF represents the delay time for groundwater outflow to reach the main channel [28]. A low value of ALPHA_BF indicates slow drainage of water from the shallow aquifer for baseflow contribution and increased storage within the aquifer. Both these parameters affect the peak flow discharges and the shape of the streamflow hydrograph.

### 2.5. Modelling to Study Impact of LULC Changes on Streamflow

The general framework adopted to study the impact of LULC change on streamflow using the SWAT model is shown in Figure 2. The hydrologic process is represented as

a function of the model parameters and initial and boundary conditions. The model parameters are functions of DEM, LULC, soil, and slope data and can be used to identify the hydrological controlling factor of the catchment. The boundary conditions correspond to the model forcings, such as rainfall, solar radiation, and other meteorological variables, at a given time step. In contrast, the initial conditions refer to the state of the variables at the start of simulations. The SWAT model simulation for the quantitative extrapolation of LULC changes on watershed hydrology and streamflow was carried out as described below:

- The watershed models were built using SWAT for two scenarios: Scenario 1: using LULC 2011; Scenario 2: using LULC 2019 (Figure 2). Except for LULC maps, all other inputs were kept invariant in both scenarios.
- The data is divided into calibration (2007–2011) and validation (2015–2019) periods (Figure 2). The model simulations of each LULC scenario were calibrated for the observed streamflow of 2007–2011.
- The sequential uncertainty fitting program algorithm (SUFI-2) algorithm in the SWAT CUP tool was used to perform the calibration, validation, and sensitivity analysis. Finally, the models were validated using the calibrated parameters. The calibrated models of the two scenarios were then simulated to identify the hydrological response under changing LULC scenarios.

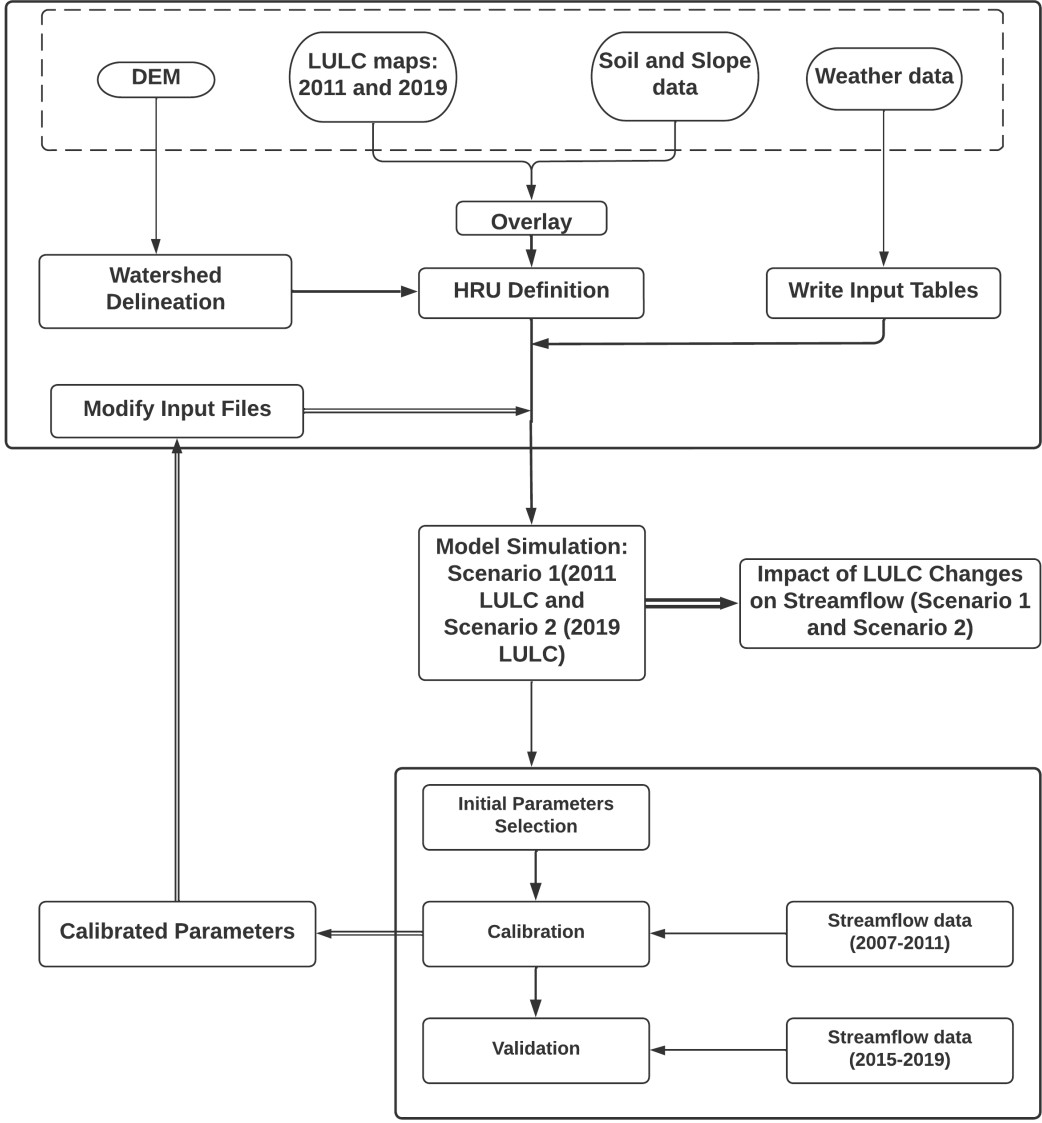

**Figure 2.** Framework for simulation and calibration of SWAT model for Scenario 1 and Scenario 2.

The variation in the relative importance of the parameter between the two scenarios was identified using global sensitivity analysis. In global sensitivity analysis, all parameter values were allowed to change and the t-stat factor and *p*-value were used to rank the parameters. The parameters with larger absolute t-stat values are more sensitive, and ones with low *p*-values (close to zero) are more significant [34].

The Nash–Sutcliffe coefficient (*NSE*) was used as a performance indicator to evaluate the hydrological prediction of the model during calibration and validation. Tan [35] suggested that, when *NSE* was used as a statistical indicator, SWAT gave 'good' model performance in identifying the occurrence of high- and low-flow. Understanding baseflow characteristics is important for studying the impact of LULC change on low-flow conditions. Several studies have also indicated *NSE* as a powerful indicator in evaluating the model performance for estimating groundwater changes [36,37]. Lee [38] recalibrated the alpha factors, reflecting the baseflow characteristics, for five watersheds using SWAT. The model performance with *NSE* as an objective function yielded a value of more than 0.6 in all the studied watersheds. Hence, *NSE* was adopted in the study. The *NSE* value varies from $-\infty$ to 1, and a value closer to 1 indicates better model performance.

$$NSE = 1 - \frac{\left(\sum_{i=1}^{n}(Q_{oi} - Q_{si})^2\right)}{\left(\sum_{i=1}^{n}(Q_{oi} - \bar{Q}_o)^2\right)} \qquad (2)$$

$Q_{oi}$ is the observed discharge, $Q_{si}$ is the simulated discharge, and $\bar{Q}_o$ is the mean of observed discharge. The model performed reasonably satisfactory during calibration and validation in both the scenarios, with *NSE* value more than 0.7.

To distinguish the impact of LULC changes on high, intermediate, and low flows, the flow duration curves (FDCs) were constructed from simulated streamflow of both scenarios. The FDC is a graphical representation of the percentage of times a given discharge is equaled or exceeded. Thus, it represents the temporal variability of the watershed's response to the rainfall-runoff process. Since it shows the distribution of streamflow over a specified time period, it portrays the impact of LULC change on the channel regime. In this study, the FDC was divided into different segments of exceedance probabilities 0–5%, 5–35%, 35–70%, 70–95%, and 95–100%, as each segment of FDC is governed by different landscape and climate processes. In both scenarios, the various segments of FDCs were analyzed to compare the impact on high- and low-flow generation.

## 3. Results and Discussion

### 3.1. LULC Change Analysis

To assess the impact of LULC changes on watershed hydrology and streamflow land-use/land-cover, maps from 2001, 2008, 2011, 2016, and 2019 were studied. The study indicated a significant change in area under urban and other land-use classes between 2011 and 2019. Hence, the LULC datasets of 2011 and 2019 were used to build two models of the watershed. The analysis showed significant changes in the area under the low-density urban residential areas (URLD), high-density urban residential areas (URHD), and urban industrial areas (UIDU), as shown in Figure 3. The areas under URHD and UIDU increased by 530 ha and 225 ha, respectively. A decline in the area under URLD (385 ha), evergreen forest (162 ha), and deciduous forest cover (132 ha) were observed between 2011 and 2019. Similarly, at the sub-basin scale, it was observed that there is a significant increase in area under URHD and UIDU, mainly at the expense of URLD, FRSE, and FRSD. In 2011, the upstream sub-watersheds had more forest cover, and urbanization was more predominant in the downstream part. However, between 2011 and 2019, in the upper part of the watershed, for example, in sub-basins 1 and 12, the expansion of URHD (7% and 12%, respectively) and UIDU (5% and 13%, respectively) occurred, with a reduction in area under deciduous (23% and 35%, respectively) and evergreen forest cover (20% and 19%, respectively). In the downstream location, similar to sub-basins 15 and 17, a loss in the area under URLD (12% and 15%, respectively) and medium density urban residential area (URMD) (10% and 6%, respectively) can be observed, with an increase in area under URHD

(6% and 15%) and UIDU (6% and 5%, respectively). The spatial distribution of change in major land-use classes is shown in Figure 4.

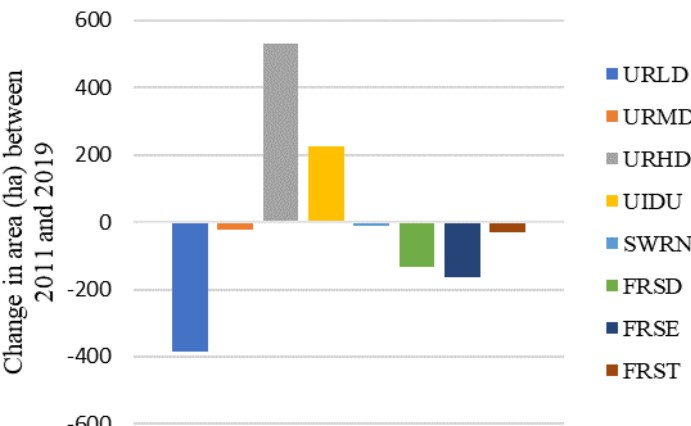

**Figure 3.** Change in the area under each LULC for the Peachtree Creek watershed between 2011 and 2019. LULC change for the watershed indicates a gain in the area under high-density developments (UIDU and URHD) and a loss in the area under low-density developments (URLD) and forests (FRSD and FRSE).

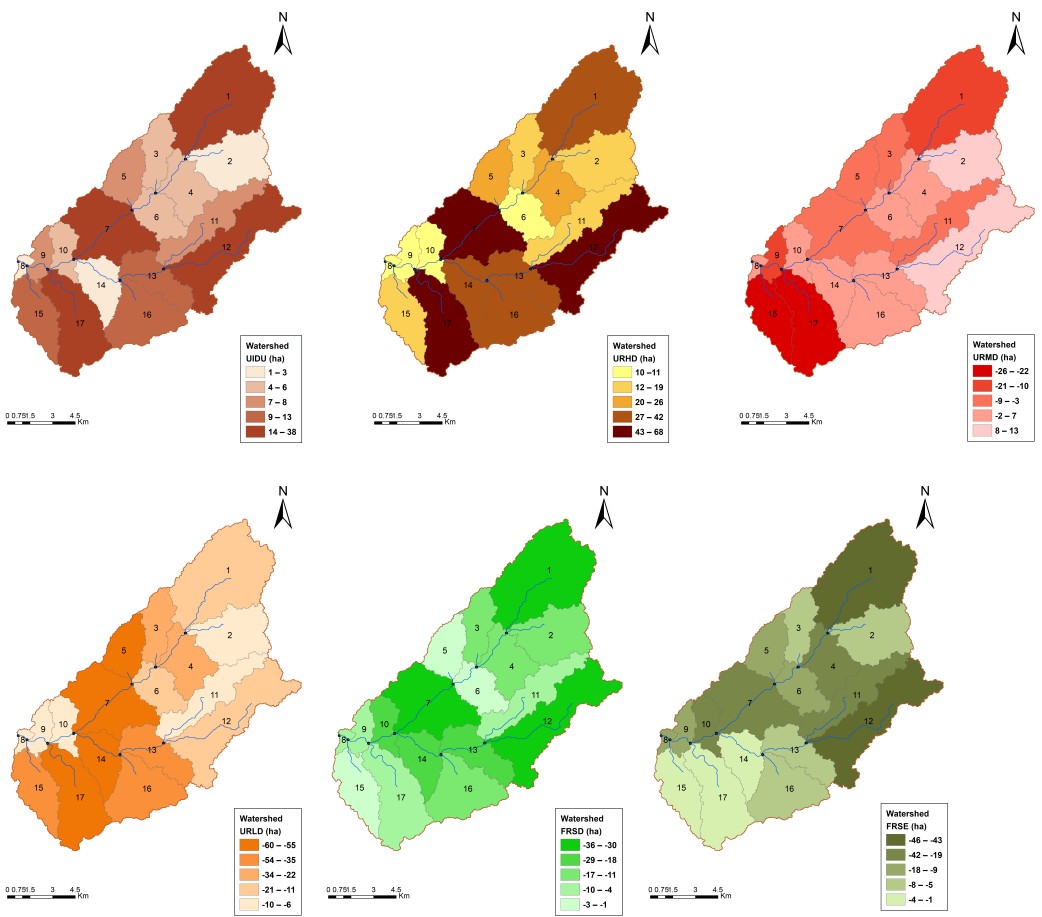

**Figure 4.** Spatial distribution of LULC change. The figure depicts net gain and loss in the area under URHD, UIDU, URMD, URLD, FRSE, and FRSD at the sub-basin scale.

### 3.2. Impact of LULC Changes on Model Parameters and Water Balance Components

To study the impact of LULC change on the hydrological regime of the basin, two scenarios were considered, Scenario 1: the model was simulated for 2011 LULC; Scenario 2: the model was simulated for 2019 LULC. The models were calibrated and validated with observed streamflows of 2007–2011 and 2015–2019, respectively. The calibrated parameters corresponding to each LULC scenario were then used to simulate the effect of changing LULC.

Figure 5a shows the percentage variation of calibrated parameters with change in the LULC scenario. The model simulations indicate an increase in the values of parameters CN2, ESCO, GW_DELAY, SOL_AWC, RCHRG_DP, and REVAPMN, whereas the values of ALPHA_BF, GWQMN, GW_REVAP, CH_K2, and SURLAG decreased in Scenario 2, compared to Scenario 1. The model parameters showing higher variations in Scenario 2 are those which influence both soil moisture and groundwater flow. Further, the lower p-value and higher t-stat value of these parameters in the global sensitivity analysis indicate higher variability to LULC changes in Scenario 2, whereas, for the entire basin, the increase in CN2 was just 1% with a change in land-cover. Figure 5b shows the percentage change in average monthly values of the water balance components for Scenario 2, with respect to scenario 1.

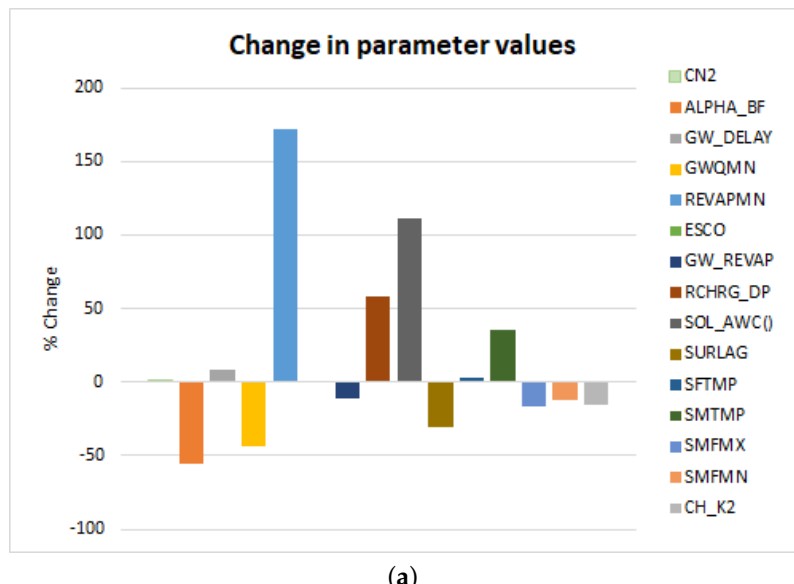

(**a**)

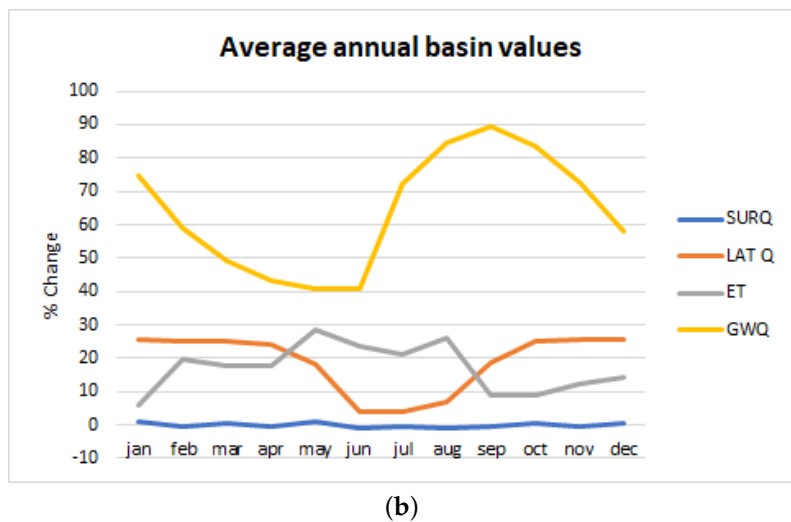

(**b**)

**Figure 5.** (**a**) Percentage change in parameter values. (**b**) Average annual WBCs for Scenario 2, with respect to Scenario 1.

**Impact on Sub-Surface Flows:**

The parameters influencing sub-surface processes, such as ALPHA_BF, SOL_AWC, and RCHRG_DP, showed variation by more than 50%. The SOL_AWC, ESCO, and GW_REVAP exhibited a higher sensitivity ranking in Scenario 2, compared to Scenario 1. The process of urbanization in the watershed resulted in the loss of forests in all sub-basins. The decreased value of GW_REVAP and increase of ESCO in Scenario 2 were due to the loss of vegetation cover, which indicates that water movement to the root zone for revap and soil's evaporative demand from lower depths reduced. The soil in the basin is primarily sandy loam belonging to hydrologic group B, characterized by moderate infiltration capacities. The higher sensitivity of SOL_AWC in post-LULC change suggests moderate soil water holding capacity and increased local warming. The variations in ESCO and GW_REVAP imply an increased baseflow contribution and lateral runoff. These parameters also affect other parameters, such as ALPHA_BF, GWQMN, and REVAPMN, in the global sensitivity analysis. The reduced values of GWQMN and the increase in REVAPMN indicate that a considerable portion of infiltered water is available for baseflow contribution. It is also observed that baseflow contribution to total flow increased by 6% in Scenario 2. However, increased SOL_AWC, GW_DELAY and imperviousness imply a higher evaporation rate and delay in the aquifer recharge. As a result, the ALPHA_BF value decreased in Scenario 2, indicating the slow drainage of baseflow from the shallow aquifer to streams. The fate of water percolating to the shallow aquifer is either lost to the atmosphere by re-evaporation or contributes to baseflow and deep aquifer recharge. The reduced revap from the shallow aquifer has resulted in increased deep aquifer recharge, as evidenced by the increased value of RCHRG_DP in Scenario 2. Thus, the results suggest an increase in baseflow and deep aquifer recharge in Scenario 2, as compared to Scenario 1.

**Impact on Surface Flows:**

A marginal increase in CN2 was observed with urbanization. This uptick in CN2 is due to the fact that URLD remains the dominant land-use in both scenarios. Hence, even though surface runoff contributes by more than 50% to total flow in both scenarios, the land-use change seems to have a negligible effect on surface runoff (SURF Q). This negligible increase in surface runoff can be attributed to increased ET. Ferguson [25] also suggested a loss of runoff in the urbanizing Peachtree Creek watershed, as a result of increased ET from remaining vegetation in urban areas, due to the advection of sensible heat from surrounding surfaces. The decreased value of the surface runoff lag coefficient (SURLAG) indicates an increase in the time of concentration. This might be due to the high-density developments in Scenario 2, with increased connectivity and density of infrastructures that can have a pronounced impact on natural runoff pathways. The decreased value of SURLAG also indicates increased flow velocities of runoff with land-use change. The land-use change of Scenario 2 is associated with a decrease in effective hydraulic conductivity of the main channel (CH_K2). Further, the study conducted by Weber [23] on the sediment budget of the Peachtree Creek watershed also indicated a decrease in hydraulic conductivity as the sediment deposition increased with urbanization.

**Impact on Evapotranspiration (ET):**

In Scenario 2, the average annual value of ET increased by 5%. Moreover, the values of parameters SOL_AWC and ESCO were higher in Scenario 2, compared to Scenario 1. This implies increased moisture loss from the soil and that the evaporative demands are met from the shallow soil layers. This will add to the ET losses in Scenario 2. Moreover, with the loss of vegetation cover, the energy input by surface radiation will increase, due to reduced albedo. The absorption of radiation by buildings and other structures in urban areas is higher than in non-urban areas. All this might add to increased local warming. Hence, in urban areas, this localized increase in surface temperature may increase the rate of evaporation. The localized warming in urban areas can lead to increased moisture loss from the bare soil, which is exposed to sunlight, due to the loss of dense forest cover, as well as from the existing vegetative surface, especially in those sub-basins having URLD as the dominant land-use.

In the present study, the variations of parameters in the urbanizing watershed indicate higher ET, reduced revap from shallow aquifer, and increased groundwater contribution to streamflow and deep aquifer recharge. The given LULC scenario seems to have a significant effect on water table dynamics and evapotranspiration, which is evident from the changes in model parameters and water balance components between the scenarios studied. However, to thoroughly analyze the different levels of impact of urbanization on streamflow, the spatial and temporal variations of the WBCs, in response to the change in the parameters must be identified, which is explained in the following section.

*3.3. Impact of LULC Changes on Streamflow*

Peachtree, being an urban watershed, streamflow remains a major portion of the water budget in both scenarios. However, the anthropic-imposed changes have resulted in the variation of the relative contribution of runoff components and evapotranspiration (ET) in Scenario 2. At the basin scale, an increase in URHD and UIDU, mainly at the expense of URLD and URMD in Scenario 2, increased evapotranspiration (ET), lateral runoff (LAT Q), and return flow to streams (GWQ). The increase in surface runoff contribution to streamflow remains negligible, whereas the portion of streamflow emanating from lateral runoff and baseflow resulted in an increase in the average annual streamflow by 9%. Consequently, stormflow conditions are impacted by the temporal and spatial variations in soil water and groundwater.

**Temporal Variations in Streamflow Due to LULC Changes:**

Figure 6 shows the monthly variation in streamflow between the two scenarios. In the winter months, due to low temperatures, ET losses are lower than in summer. However, the percentage increase in ET is higher, even during the winter months (Dec–Feb), in Scenario 2, as compared to Scenario 1. This increased evaporation might be due to the local warming of urban areas because of increased anthropogenic activities. This reduces the availability of snow for melting and increases soil moisture deficit. This is evident from the variation in snow parameters (SMTMP, SFTMP, SMFMX, and SMFMN), which suggests a lower melting rate. Moreover, the soil permeability is reduced due to low temperatures. Hence, aquifer recharge is reduced. This lowers baseflow contribution and reduces streamflow in the winter months by 2–4% in Scenario 2 (Figure 6). A similar trend of decline in percentage increase in GWQ occurred in spring (Mar–May) and fall (Sept–Nov) seasons, as well. This reduction in GWQ in all these months can be due to the low temperature and reduced permeability of soil, thus lowering the shallow aquifer recharge. However, increased LATQ and reduced ET losses compensate for the decrease in GWQ. Hence, the percentage decrease in streamflow is lower in winter, as compared to summer months.

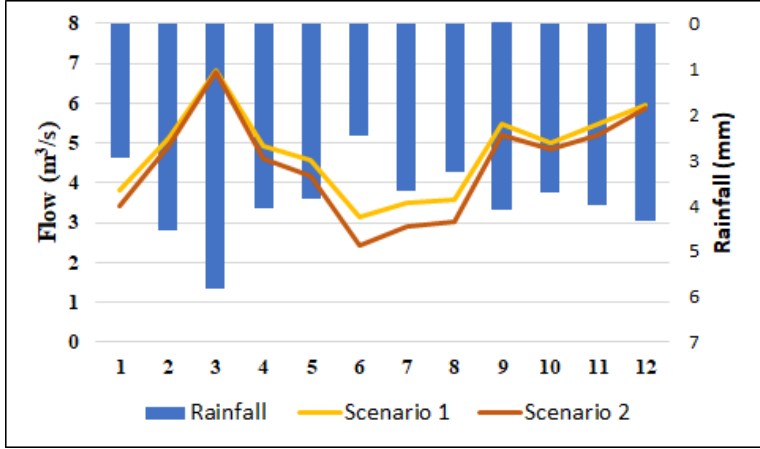

**Figure 6.** Monthly average values of streamflow (for scenario 2 and scenario 1) and rainfall.

In accordance with the variation in runoff components and ET in Scenario 2, a higher decrease in the stream flow is observed during the summer months (June-August) by 10–25% (Figure 6). This is because the monthly variations of the water balance components indicate that the percentage increase in ET is higher (21–28%) during the summer for Scenario 2. During the same season, LAT Q decreased with an increase in ET. This reduction in shallow sub-surface flow can be due to increased moisture loss and the drying up of soil. A slight reduction of surface runoff is also observed during the summer months, due to increased surface warming, as well as evaporation, due to urbanization. Baseflow contribution to streamflow (GWQ) is lowest in the month of June. In the same month, the streamflow was reduced by 25% (Figure 6). The baseflow became dominant toward the late summer. This is evident in Figures 5b and 6, as the streamflow starts rising, despite the reduction in LATQ. Watson [39] also observed an increase in GWQ towards late summer because the water stored in the shallow aquifer during the winter and summer months starts contributing to baseflow. Since scenario 2 is characterized by the slow drainage of water from a shallow aquifer, groundwater outflow to the stream channel appears towards late summer.

**Spatial Variations in Streamflow Due to LULC Changes:**

Figures 7 and 8 depict the sub-basin scale variation in WBCs and streamflow variation, respectively. To assess the impact of individual land-use change and its spatial scale effect, sub-basin scale assessment was performed. For example, consider sub-basins 1 and 17, located at upstream and downstream parts of the watershed, respectively, having UIDU as major land-use class. The LULC change scenario of SB 1 is associated with the conversion of forest to urban areas, whereas in SB 17, the low density and medium density residential areas are converted to UIDU and URHD. Both these sub-watersheds have decreased groundwater recharge and increased ET in Scenario 2, due to higher impervious fractions. This lowers the water table and reduces the baseflow contribution to streams. The increase in deep aquifer recharge in these sub-basins also suggests that water from shallow aquifer percolates to the deep aquifer and is lost from the system. Thus, in watersheds with UIDU as the major land-use class, further urbanization eliminated baseflow contribution to streams and streamflows emanates from surface runoff and lateral flow. Hence, LULC changes in these watersheds resulted in decreased streamflow (0–7%) (Figure 8). The sub-basins with URLD as the dominant land-use class, for example, sub-basin 12, also experiences a reduction in groundwater recharge and increased ET with urbanization. However, the percentage decrease in recharge is lower than those sub-basins with UIDU as the major land-use. The water-replenishing shallow aquifer contributes to baseflow to streams and for deep aquifer recharge. The partition of streamflow indicates that baseflow has more contribution to total flow than LAT Q. Thus, these sub-watersheds have increased streamflow (15–30%) in Scenario 2 (Figure 8).

The sub-basin scale study indicates that high-density developments lead to shift from baseflow dependence to surface runoff and lateral flow dependence, whereas the sub-basins with a majority of area under low-density developments, urbanization resulted in increased streamflow, due to higher baseflow contributions. Hence, the spatial scale analysis indicated that urbanization has significant effect on baseflow and that streamflow variability is a function of both the degree of urbanization and predominant land-use class of the sub-watersheds.

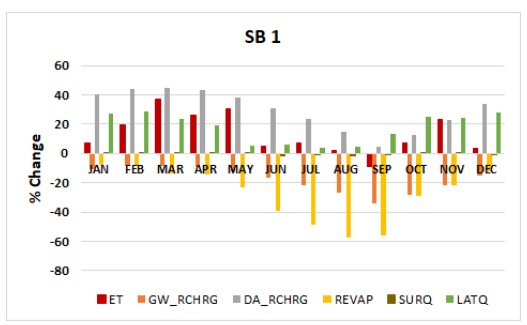
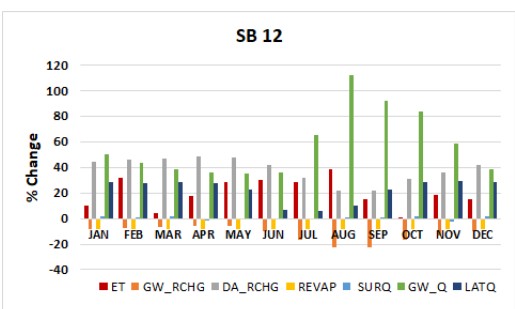
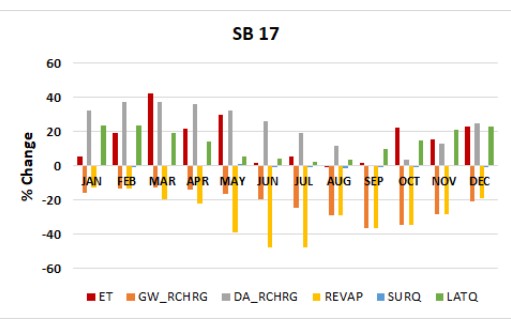

**Figure 7.** Percentage change in WBCs for sub-basins 1, 17, and 12—comparison between scenario 1 and scenario 2 with change in LULC.

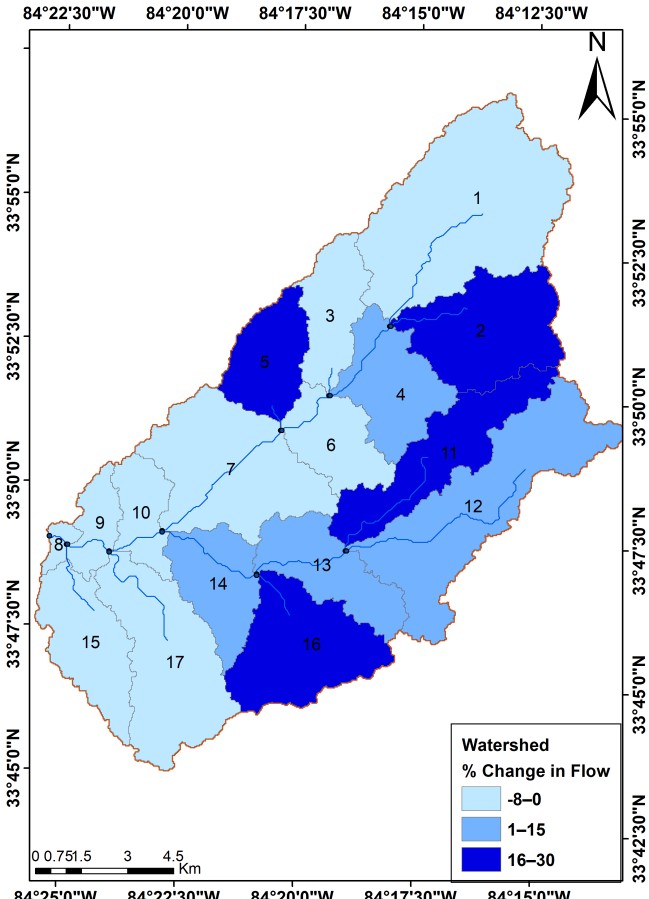

**Figure 8.** Percentage change in streamflow at the sub-basin scale between scenario 1 and scenario 2. The sub-basins with URLD as major LULC exhibit an increasing trend in streamflow (darker blue shades), whereas those sub-basins with UIDU as major LULC (light blue shade) show a decreasing trend in streamflow.

### 3.4. Impact of LULC Changes on High and Low Flows

The study indicated that the rate of groundwater contribution to streams is highly modified with urban developments. As seen from the temporal scale analysis (Figure 5b), the baseflow contribution to streams is highly susceptible to annual variations in temperature and precipitation. This has resulted in a significant reduction of streamflow during dry periods. It is evident from the sub-basin scale study that most of the sub-watersheds located in upstream and downstream have decreased baseflow contribution with LULC changes. Hence, the streams in these sub-basins are vulnerable to reduced low flows during the dry period. An increase in streamflow can also be observed in some sub-basins located upstream, due to increased baseflows. Since the LULC change scenario has a negligible effect on surface runoff, the increased baseflow contributions have the potential to make the sub-watersheds vulnerable during flood events by increasing discharge to streams, resulting in higher peaks at downstream. The results show that spatial and temporal variations in both low and high flows are affected by anthropic-induced land-use alterations.

Flow duration curves can be visualized as a statistical representation of the portioning of runoff components due to weather and catchment properties. From FDC (Figure 9), it can be observed that there is approximately 5–10% increase in high flows (0–5% probability exceedance), whereas the low flows are reduced by 65–69% (95–100% probability exceedance). The comparative analysis suggested that land-use change the reduced intermediate and low flows, thus having a negative impact on long-term sustainability of flows in the absence of precipitation. The study conducted by Peters [22], in comparing the streamflow response of highly urbanized Peachtree Creek watershed with streams in less urbanized areas of Atlanta region, also indicated a significant increase in high flows and a decrease in low-flow values, as compared to streams in less urbanized watersheds.

In Scenario 2, a decrease in the pervious fraction, with an increase in URHD and UIDU, at the expense of URLD and vegetation cover, might result in infiltration excess overland flow. This is evident from the slight uplift of the high-flow segment, especially in the range of 0–5 exceedance percentages, which represents fast flow (surface or overland flow) response. This fast flow response during large precipitation events, coupled with increased baseflow due to LULC changes, results in greater discharge to streams. Thus, urbanization of the watershed results in the increased magnitude and frequency of peak discharges. The comparative analysis of intermediate and low-flow segments of FDCs of the two scenarios suggest that slow-flow components of runoff are highly sensitive to land-use changes. The behaviour of intermediate and low-flow segments to LULC changes is highly correlated to intra-annual variations in climatic conditions. The middle and lower parts of FDC depicts the flow regime of the channel corresponding to extended dry periods or low rainfall. The FDC was downshifted in Scenario 2 for the exceedance probability greater than 70%. Scenario 2 is characterized by increased ET and delayed drainage of groundwater outflow, and this will be further exacerbated during dry periods, leading to reduced sub-surface water storage and baseflow contribution. Thus, downshift of FDC might be a consequence of both reduced baseflow in dry periods and increased evaporation losses due to anthropogenic heat emissions. It is observed that sustained periods of low flows are greater with Scenario 2 than in Scenario 1.

Under the same climatic conditions, the shape of FDCs generated in both scenarios can be used to identify the runoff variability driven by land-use change. Above 70% probability exceedance, the slope of the curve is relatively less steep in Scenario 2, as compared to Scenario 1. Thus, the shape of the FDCs in both scenarios indicates that land-use alterations have a moderate impact on fast flow components of runoff, whereas the intermediate flows due to soil storage and low flows due to deep groundwater flows are affected. The downshift of flow duration curve with 2019 LULC is the combined effect of surface and sub-surface responses to changes in water and energy fluxes, due to anthropic-induced changes.

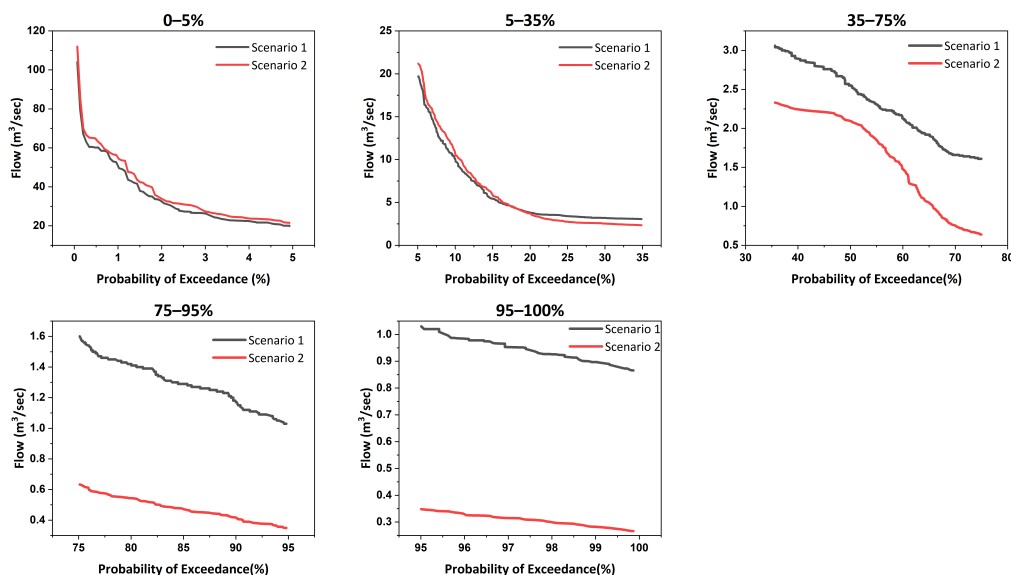

**Figure 9.** Flow duration curve is divided into five segments, based on the percentage of exceedance frequency, to evaluate the occurrence of high flows (0–5% and 5–35%), intermediate flows (35–70%), and low flows (70–95% and 95–100%).

## 4. Conclusions

The study assesses the impact of anthropic-induced land-use/land-cover (LULC) changes by quantifying the hydrological modifications induced in the watershed and by analyzing streamflow variability. The study was conducted in the urbanizing watershed of Peachtree Creek. The variations in model parameters with change in LULC characteristics indicates the impact on hydrological components. The interaction between the change in WBCs and streamflow variations at the sub-basin scale showed different levels of impact due to urbanization. This helps in comprehending the influence of the predominant land-use class on the fundamental hydrological process. Streamflow variations at monthly time scale were studied to analyze the hydrological regime of the basin and their quantitative reduction during intra-annual seasonal variation in response to LULC changes. The variations in the FDCs were studied to evaluate the flooding vulnerability and the ability to sustain low flows under changing LULC scenarios in the catchment. The findings are concluded as follows:

- The variations in model parameters suggest that the LULC changes of Scenario 2 show higher sensitivity to parameters controlling sub-surface flows and evapotranspiration losses.
- The changes in WBCs indicate higher ET, reduced revap from shallow aquifer, and increased groundwater contribution to streamflow and deep aquifer recharge. However, the surface runoff shows minor variation, with LULC alteration due to increased evaporation losses.
- The spatial scale assessment indicated that water available for baseflow depends on the dominant LULC of the sub-basin. For instance, the results showed that there is a reduction in streamflow for those sub-basins with UIDU as a major land-use class, whereas increased streamflow in basins with URLD as the dominant land-use. This is because the former has increased the percentage of impervious fractions, resulting in negligible groundwater contribution and increased urban ET.
- Total discharge to streams significantly reduced during early summer (especially in June), due to the slow drainage of water from the shallow aquifer to streams and increased ET. On the other hand, towards late summer, the groundwater contribution to streamflow increases (i.e., the water stored in the aquifer during the winter and spring).

- The occurrence of extreme flows was analyzed using FDCs, representing the flow regime as a function of intra-seasonal variation in climate factors and changes in catchment characteristics in response to LULC alterations. As expected, the results show an increase in peak flow and reduced low-flow due to urbanization. Further, the FDCs in Scenario 2 are characterized by flatter curves, compared to Scenario 1, reflecting an increase in sub-surface flows.
- The upper part of FDC relates to the streamflow during heavy precipitation events. During the wet period, when energy is limiting, the portion of streamflow emanating from lateral runoff and baseflow increases. This addition of sub-surface flows leads to an uplift of high-flow segment, indicating an increased magnitude and frequency of peak discharge with urbanization. This, in turn, increases vulnerability to floods during periods of intense rainfall. Moreover, higher peak flows and decreased value of effective channel hydraulic conductivity (CH_K2) with urbanization may lead to reduced channel stability and increased sediment and pollutant loading.
- The behavior of intermediate and low-flow segments of FDCs is mainly governed by sub-surface storage and slow-flow components (baseflow). The downshift of the middle and lower parts of FDCs in Scenario 2 indicates an increase in ET losses, delay in the generation of groundwater outflow, and increased baseflow days, resulting in a downshift of the middle and lower parts of FDCs. Thus, it shows that urbanization has a negative impact on the long-term sustainability of flows in the absence of precipitation.
- Thus, LULC changes due to anthropic activities (especially urbanization) indicate an increased susceptibility to floods during heavy precipitation events and reduced reliability of streams during the dry period.

The present study shows that variations in streamflow are a function of the degree of urbanization, LULC class undergoing transition, and predominant land use of the sub-watershed. It is to be noted that the changes in model parameters in response to LULC alterations can provide valuable insight into the hydrological processes controlling streamflow variation and high- and low-flow generation. It was observed that changes in the flow regime of the channel may not always be a consequence of variation in the surface runoff but depend on the ET and groundwater dynamics. It can also be concluded from the study that ET plays a major role in determining the portioning of precipitation in urban water balance. This is because LULC changes alter the net radiation available, land surface roughness, and transpiration changing the magnitude and rate of ET. The sub-surface water storage and groundwater outflow to streams show high spatial and seasonal variations, depending on evapotranspiration loss. This leads to water surplus/deficit regions and the generation of extremes.

The study attempted to assess how changes in model parameters and their sensitivity can solely represent the changes in hydrological processes and streamflow, in response to LULC changes. The study outlines the significance of identifying the spatial and temporal scale variation in streamflow help in prioritizing the sub-basin and time period for implementing management activities to prevent over-exploitation of land and water resources. Knowledge about the changes in high- and low-flows helps in mitigating the impacts of floods and drought. However, integrating the coupled effect of climate and LULC in modeling a watershed's hydrologic processes can better predict the occurrence of hydro-climatic extremes. Further, uncertainty regarding the areal locations, frequency, and magnitude of extreme flows under the changing environment requires further attention.

**Author Contributions:** Conceptualization, R.S.; methodology, U.R.P. and R.S.; software, U.R.P.; validation, U.R.P. and R.S.; formal analysis, U.R.P. and R.S.; investigation, U.R.P. and R.S.; resources, R.S.; data curation, U.R.P.; writing—original draft preparation, U.R.P. and R.S.; visualization, R.S. and A.S.; supervision, R.S.; project administration, R.S. and A.S.; funding acquisition, R.S. and A.S. All authors have read and agreed to the published version of the manuscript.

**Funding:** This research received no external funding.

**Data Availability Statement:** Not applicable.

**Acknowledgments:** The authors would like to express sincere thanks to the editor and anonymous reviewers for constructive feedback and comments. The first author is grateful to the HydroSystems Research group, IIT Tirupati for their support.

**Conflicts of Interest:** The authors declare no conflict of interest.

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
