# Peer review of "Multiscale Variability of Hydrological Responses in Urbanizing Watershed"

_remotesensing, doi:10.3390/rs15030796_

Round 1
Reviewer 1 Report
Dear authors:
I made some comments in the file, which is attached. Considering the scenarios, my main concern is how the calibration process was conducted. To me, this part of the paper is very confusing. In the Introduction, there is excessive text regarding the SWAT's parameters and just a few for LULC using this model. I recommend improving the Introduction with highlighted papers on this subject.
The results are good, but you need to examine how and if the SWAT for LULC makes sense in the way it was used. I have serious concerning here.
Good luck.

Author Response
We appreciate the reviewer's insightful comments and recommendations. All of the comments have now been addressed in the revised manuscript. In the file attached, all points mentioned are addressed.

Reviewer 2 Report
The paper is titled “Multiscale Variability of Hydrological Responses in Urbanizing Watershed.” The purpose of this article is to quantify the induced hydrological changes in an urbanizing catchment and to analyze extreme flow variability using a soil and water assessment (SWAT) tool for Peachtree Creek, United States.
The article is interesting, but needs improvement:
- the introduction needs to be shortened because it is too long. The introduction takes over 3 pages of text. Some of the literature cited in the introduction is not directly applicable to the problem discussed in the text of the article. Detailed information on the methodology should be included in the discussion.
- the research area was not well described. Basic information on the analyzed catchment is missing. The map presenting the research area does not bring anything. Fig 1 does not give the non-US reader basic knowledge: where exactly the research area is located, what are the heights of the terrain, geological structure, what is the land use, groundwater level. The mere mention of the research area in lines 200-221 is insufficient. In addition, the authors do not provide information whether systematic hydrological observations are carried out on a given watercourse or not. If there are flow measurements, they should be discussed, if not, indicate how they were calculated. In fig. 2 there is information that data on observed flows and flows 2007-2011 and 2015-2019. It's unclear to me. The study area was divided into sub-basins, and each of these sub-basins should be properly characterized (e.g. area, land development, etc.)
- the article lacks a discussion of how comparable the results used are with other studies
- there should be more literature items in the article, especially after 2020.
The article should be written in such a way that any interested reader would be able to recreate the proposed methodology, as well as indicate the advantages and disadvantages compared to other methods.
Author Response

(The authors gave the same response as above.)

Reviewer 3 Report
I reviewed the manuscript "Multiscale Variability of Hydrological Responses in Urbanizing Watershed". The paper aimed to quantify the hydrological changes induced in the urbanizing watershed and changes in streamflow variability and generation of extremes (high and low flows), using the Soil and Water Assessment Tool (SWAT). The ms. is well done but I would like to make some minor suggestions:
1. Could the authors insert "climate changes" as keywords too?
2. Figure 1. I suggest including the localization inside the country too. Remember that this journal has readers from every world.
3. Figure 2a. It is difficult to read (small letters). Please, improve.
4. I suggest inserting some discussion about the impact of climate change on the hydrological responses. This is important to improve the discussion and get the attention for the readers and water managers.
Author Response

(The authors gave the same response as above.)

Round 2
Reviewer 1 Report
Dear authors:
Thank you for taking my comments into account. I believe the paper is in good shape.
Congratulations!
Author Response
We are thankful to the reviewer for the time and valuable comments on the paper
Reviewer 2 Report
In my opinion, the authors only partially improved the article. My comments are as follows:
- the authors did not use the option to track changes in the article. The new text has been marked with a different font color, but you can't see what has been deleted in the text
- the number of literature items is poor
- fig. needs improvement, select the US outline, the Georgia outline should contain more elements on the basis of which readers will visually identify the research area. In its current form, it is impossible without checking the geographic coordinates
- the list of references should be adapted to the requirements of the journal e.g. line 711-712
- the methodology was not well explained
- there was no discussion of the results obtained
Author Response
We appreciate the reviewer's helpful comments and suggestions. We responded to all concerns and sent two additional files with the response letter that highlighted the track changes in the article done in rounds 1 and 2. Please see the attached document.
